# Is Breast Cancer Risk Associated with Menopausal Hormone Therapy Modified by Current or Early Adulthood BMI or Age of First Pregnancy?

**DOI:** 10.3390/cancers13112710

**Published:** 2021-05-31

**Authors:** Eleni Leventea, Elaine F. Harkness, Adam R. Brentnall, Anthony Howell, D. Gareth Evans, Michelle Harvie

**Affiliations:** 1Department of Nutrition and Dietetics, Cambridge University Hospitals NHS Foundation Trust, Cambridge CB2 0QQ, UK; eleni.leventea@addenbrookes.nhs.uk; 2Nightingale Breast Screening Centre & Prevent Breast Cancer Unit, Wythenshawe Hospital, Manchester University NHS Foundation Trust, Manchester M23 9LT, UK; Elaine.F.Harkness@manchester.ac.uk (E.F.H.); Anthony.Howell@manchester.ac.uk (A.H.); Gareth.Evans@mft.nhs.uk (D.G.E.); 3Centre for Imaging Science, Division of Informatics, Imaging and Data Science, Faculty of Biology, Medicine and Health, University of Manchester, Manchester Academic Health Science Centre, Manchester M23 9LT, UK; 4NIHR Manchester Biomedical Research Centre, Manchester M13 9WU, UK; 5Centre for Cancer Prevention, Wolfson Institute of Preventive Medicine, Barts and The London School of Medicine and Dentistry, Queen Mary University of London, London EC1M 6BQ, UK; a.brentnall@qmul.ac.uk; 6Manchester Breast Centre, The Christie Hospital, Manchester M23 9LT, UK; 7Division of Cancer Sciences, Medicine and Health, University of Manchester, Manchester Academic Health Science Centre, Manchester M23 9LT, UK; 8Manchester Centre for Genomic Medicine, Manchester University Hospitals NHS Foundation Trust, Manchester M23 9LT, UK; 9NW Genomic Laboratory Hub, Manchester Centre for Genomic Medicine, Manchester University Hospitals NHS Foundation Trust, Manchester M13 9WL, UK; 10Faculty of Biology, Division of Evolution and Genomic Sciences, School of Biological Sciences, Medicine and Health, University of Manchester, Manchester Academic Health Science Centre, Manchester M23 9LT, UK

**Keywords:** menopausal hormone therapy, breast cancer risk, BMI, early BMI, age of pregnancy

## Abstract

**Simple Summary:**

Menopausal hormone therapy (MHT) increases risk of developing breast cancer (BC), and women are often advised to avoid its use for this reason. In this analysis we examined the size of this effect using data from a large cohort of women attending breast cancer screening in Manchester, UK. We additionally explored the extent to which risk from MHT might be modified by current BMI, early adulthood body mass index (BMI) (age 20 years), and age of first pregnancy. Identifying modifying effects would help enable better estimation of risk associated with MHT for an individual woman. Results indicated that women using combined oestrogen and progestagen MHT were at greater risk than those receiving oestrogen-only MHT. The Relative risk associated with MHT was less for obese women than non-obese women. After adjustment for current BMI, the effect of MHT did not appear to be substantially modified by early BMI or age of pregnancy.

**Abstract:**

Menopausal hormone therapy (MHT) has an attenuated effect on breast cancer (BC) risk amongst heavier women, but there are few data on a potential interaction with early adulthood body mass index (at age 20 years) and age of first pregnancy. We studied 56,489 women recruited to the PROCAS (Predicting Risk of Cancer at Screening) study in Manchester UK, 2009-15. Cox regression models estimated the effect of reported MHT use at entry on breast cancer (BC) risk, and potential interactions with a. self-reported current body mass index (BMI), b. BMI aged 20 and c. First pregnancy >30 years or nulliparity compared with first pregnancy <30 years. Analysis was adjusted for age, height, family history, age of menarche and menopause, menopausal status, oophorectomy, ethnicity, self-reported exercise and alcohol. With median follow up of 8 years, 1663 breast cancers occurred. BC risk was elevated amongst current users of combined MHT compared to never users (Hazard ratioHR 1.64, 95% CI 1.32–2.03), risk was higher than for oestrogen only users (HR 1.03, 95% CI 0.79–1.34). Risk of current MHT was attenuated by current BMI (interaction HR 0.80, 95% CI 0.65–0.99) per 5 unit increase in BMI. There was little evidence of an interaction between MHT use, breast cancer risk and early and current BMI or with age of first pregnancy.

## 1. Introduction

Approximately 80% of women going through the menopause experience symptoms [1]. Menopausal hormone therapy (MHT) is the most effective treatment option. MHT use halved in the early 2000s as a result of widely publicised associations between MHT use and increased risk of breast cancer and thromboembolism. Rates stabilised in the 2010s and currently there are an estimated one million MHT users in the UK each year, representing approximately 10% of women passing through the menopause [1]. A recent meta-analysis based on 58 prospective and retrospective studies, including 568,859 women and 143,887 breast cancer cases concluded that ever MHT use is associated with increased breast cancer risk (RR 1.26, 95% CI 1.24–1.28). Risk is higher for current compared to past users and increased with longer MHT use. Amongst current users, risk is greater with combined (oestrogen plus progestagen) MHT (RR for 5–14 years of use 2.08, 95% CI 2.02–2.15) compared to oestrogen only MHT (RR 1.33, 95% CI 1.28–1.37). Risk was attenuated amongst heavier women, particularly for oestrogen-only MHT, with little additional risk from oestrogen-only MHT in women who were obese [2]. 

MHT use increases breast cancer risk amongst women in the general population but also in women with increased risk due to familial cancer [3]. Many women, especially those at higher risk of breast cancer, are counselled to avoid or minimise MHT use [4]. Guidelines recommend an individual risk benefit assessment for prescribing MHT [5]. However there are limited data on whether MHT risk is modified by other patient characteristics which would help to inform this decision. 

Increased body mass index (BMI) after the menopause (RR per 5 BMI units: 1.12, 95% CI 1.09–1.15) and weight gain throughout adult life after the age of 20 (RR per 5 kg: 1.06, 95% CI 1.05–1.08) are consistently associated with increased risk of postmenopausal breast cancer [6]. In contrast, higher weight during adolescence and early adulthood (age ≤30 years) are observed to have an inverse effect [7]. High body adiposity in early adulthood is associated with a reduced risk of postmenopausal (RR per 5 BMI units: 0.82, 95% CI 0.76–0.88) and premenopausal breast cancer (RR per 5 BMI units: 0.82, 95% CI 0.76–0.89) [6]. We recently reported higher BMI in early adulthood (>23.4 kg/m^2^) negated the impact on risk of high later attained BMI [8].

Late age of first pregnancy (after the age of 30 years) and nulliparity are associated with increased risk of breast cancer. Women with first pregnancy after the age of 30 have approximately twice the risk of developing breast cancer compared to women with first pregnancy before the age of 20, and nulliparous women have a 30% increased risk compared to parous women [9]. Risks associated with attained adult and early adulthood weight and age of pregnancy are all in part mediated by different exposure to oestrogen and progesterone, and may alter the hormone responsiveness of the breast [10]. Thus they may modify the risk associated with MHT use.

Here we sought to examine the association between combined and oestrogen only MHT use and breast cancer risk in a cohort of women from the National Health Service Breast Screening Programme (NHSBSP), in Greater Manchester, UK. We examined whether BC risk associated with these types of MHT are modified by BMI at study entry (age 46–84 years) and/ or early adulthood BMI (age 20), or age at first pregnancy or nulliparity.

## 2. Methods 

### 2.1. Population 

The Predicting Risk of Cancer At Screening (PROCAS) study has been described in detail elsewhere. In total 57,902 women aged between 46 and 84 years in the National Health Service Breast Screening programme (NHSBSP) were recruited from five areas of Greater Manchester (Manchester, Oldham, Salford, Tameside and Trafford) between October 2009 and June 2015 [11]. Recruitment was carried out in two phases: initially all women who were invited for three-yearly breast screening (October 2009–October 2012) after which only women invited to their first screen in the area (mainly aged 46–53 years) were invited to participate in the study. Participants were invited once during the recruitment period. During the initial phase uptake to screening was 68% with uptake to PROCAS 37% of attendees, in the second phase screening uptake was 58% and uptake to PROCAS was 47% of attendees (screening uptake is lower in first time invitees). 

### 2.2. Data Collection

Data collection was based on a two-page questionnaire, which was sent to participants with a consent form between their invitations to attend screening and scheduled screening appointment (Appendix A).

The self-reported questionnaire gathered information on risk factors for breast cancer including: previous breast cancer diagnosis, breast or ovarian cancer in first and second degree relatives, hormonal risk factors, including age at menarche, oophorectomy and hysterectomy, menopausal status, age at menopause, parity and age at first pregnancy, physical activity levels, alcohol intake and ethnicity. Questions in relation to MHT use included name of preparation, duration of use and when MHT was last used (if no longer on MHT). Women were also asked to record their height, current weight and recalled weight at age 20 as a proxy of early adulthood BMI. Current BMI and BMI at 20 were calculated from these variables. Completed questionnaires were collated by the study team and entered into the study database. 

### 2.3. Diagnosis of Breast Cancer

The primary outcome was diagnosis of a new breast cancer (invasive or ductal carcinoma in situ), from entry to PROCAS onwards, as identified through the NHSBSP and the Somerset and North West Cancer Intelligence services. Follow-up (median eight years) was censored at date of breast cancer diagnosis, date of death, date lost to follow-up, e.g., moving out of the area, or date cancer databases were last checked (April 2020). The current analysis excluded women with breast cancer diagnosed prior to study entry (*n* = 895). 

### 2.4. MHT Use

MHT use was classed as never, current and former. Never users were women who indicated they had never been on MHT at any time. Current users were women who reported they were still on MHT and gave details about length of time on MHT but did not indicate a time of stopping. Former users were women who reported no longer using MHT but indicated how long they had been using MHT and time since stopping. 

Where women did not provide an MHT name, it was assumed that those who had a hysterectomy had oestrogen only MHT, whereas women who did not have hysterectomy had combined (oestrogen plus progestagen) MHT. MHT status was missing for 514 women who were excluded from any further analyses (Figure 1). 

The analyses included pre/peri and postmenopausal women enrolled in the PROCAS study. Women were considered postmenopausal if they indicated on the questionnaire they had been through the menopause (i.e., not had period for 12 months) or reported they had both ovaries removed (surgical menopause) or current use of MHT or age at menopause was unknown but age at time of questionnaire completion was 55 years or over as based on criteria defined by Phipps et al. [12].

### 2.5. BMI Data 

We excluded attained BMI and early BMI values which were >60 or <16 kg/m^2^. For current BMI, 3943 were unknown or beyond these cut-offs. For early BMI, 6300 values were unknown or beyond these cut-offs. Median value of the cohort was assumed for missing values (total 6.8% for current and 10.9% for BMI at age 20).

### 2.6. Statistical Analysis 

#### Demographic Characteristics and Breast Cancer Risk Factors amongst MHT Users and Non-Users

Differences in demographic characteristics and breast cancer risk factors across MHT groups (never, current, former) were tested using one-way ANOVA and Chi-square tests where appropriate. Adjusted comparisons of continuous measure used linear regression (with covariates for age (years), current and early BMI ( kg/m^2^)).

### 2.7. MHT Use and BC Risk 

Cox (or proportional hazards) regression was used to model the relationship between MHT use and diagnosis of breast cancer. Follow-up was censored at date of breast cancer, date of death or date of last follow-up (April 2020). Results were expressed as hazard ratios (HR) and 95% Wald confidence intervals (95% CI).

Fully adjusted Cox regression models included the following established risk factors for breast cancer: age (1 year), height (5 cm), BMI (5 units), early BMI (5 units), ethnicity (white/other), age at menarche (1 year), age at first pregnancy (<20, 20–24, 25–29, 30–34, ≥35) years, parity, age at menopause (1 year) menopausal status (pre/peri or postmenopausal), oophorectomy, self-reported exercise (1 h/week) and alcohol (1 unit/week), MHT status (current, former, never), MHT type (combined, oestrogen only) and family history (first or second degree). For the fully adjusted analysis, missing values were imputed as the median value of the cohort. We did not include the available mammographic density data in the models since there is some evidence this may be part of the pathway for reduced risk alongside higher early BMI [13]. The multiple deprivation score was not included since this was not associated with risk once the variables associated with this, such as age of pregnancy were included in the model (Appendix A).

### 2.8. MHT Use, Breast Cancer Risk and Effect Modification by Current BMI and Early BMI and Age of First Pregnancy

To assess whether current and early BMI modified the relationship between MHT and the risk of breast cancer the following two way interaction terms were also included in the fully adjusted models: current/ former MHT use* current BMI * (Appendix A) and current/ former MHT use* early BMI (Appendix A). We also tested the 3-way interactions: current/ former MHT use*current BMI *early BMI (Appendix A).

For presentation the relationship between MHT use and BC risk was tabulated by stratifying above and below the median for current (26.4 kg/m^2^) and early (21.6 kg/m^2^) BMI. The reference group was never use of MHT, current BMI below the median and early BMI below the median. The median was used to dichotomise results and aid with interpretation (Table 1). 

The relationship between late age at first pregnancy and nulliparity, MHT status and BC risk was assessed using analysis stratified by age at first pregnancy <30 years or a combined group which included women with first pregnancy ≥30 years or who were nulliparous (Table 2 and Appendix A).

Due to previous known heterogeneity by type of MHT and subtype of breast cancer we also performed these analyses according to (i) MHT type (combined oestrogen and progestagen or oestrogen only) (Table 3) and (ii) oestrogen receptor positive breast cancer. (Appendix A).

## 3. Results

### 3.1. Flow Diagram

From the cohort of 57,902 women, 895 were diagnosed with breast cancer prior to study entry, 4 were lost to follow up and were excluded from the analysis. Women with unknown MHT use status were also excluded from the analysis (*N* = 514). The denominator for the main analysis was 56,489 with the endpoint being a new diagnosis of breast cancer (*N* = 1663, 3.2% of the cohort) (Figure 1).

### 3.2. Demographic Characteristics and Breast Cancer Risk Factors According to MHT Use

Demographic characteristics and BC risk factors according to MHT groups are shown in Table 4. Of those eligible for analysis 7.8% were current MHT users, 28.6% were former users and 63.6% had never used MHT. Across all MHT groups at study entry, 35.9% were in the underweight or normal BMI category range, 39.5% in the overweight, 24.6% in the obese BMI category, 3801 women had an unknown BMI. The median age at first pregnancy was 24 years (interquartile range [IQR] 21–28 years), 26.7% of women had their first pregnancy either at or after the age of 30 years. In total, 27.9% of women had a first or second degree family history of ovarian and/ or breast cancer. 

When compared to never users, current MHT users had a higher percentage of oophorectomy (27.5% vs. 5.6%), lower BMI at study entry (median: 26.0 vs. 26.4 kg/m^2^), were less likely to have a family history of ovarian and/ or breast cancer (26.6% vs. 28.3%), a lower deprivation score (median: 17.6 vs. 19.4), and younger age of first pregnancy (median 23 vs. 24 years). As expected, former users were the oldest group, also reflected by the higher percentage of participants being postmenopausal. 

Compared with former users, current users were younger (median: 55.2 vs. 62.9 years), had longer use of MHT (median: 7 vs. 5 years), higher percentage use of oestrogen only MHT (51.7% vs. 41.0%), lower BMI at entry (median: 26.0 vs. 26.4 kg/m^2^), were less likely to have family history of breast and/ or ovarian cancer (26.6% vs. 27.1%), and lower deprivation score (median: 17.6 vs. 18.7). All differences cited have *p* < 0.001 due to large sample size. Women with known MHT status were lighter at entry to PROCAS, more likely to be white, and from less deprived backgrounds than those with unknown MHT status (Appendix A).

### 3.3. MHT Use Status and Breast Cancer Risk 

Median follow up from study entry was 8 years (IQR: 7–9, minimum 5 and maximum 12 years). Compared with never use, current MHT use was positively associated with breast cancer (HR 1.35, 95% CI 1.13–1.60), while there was little evidence of an association for former users (HR 1.03, 95% CI 0.91–1.17) (Table 5). The fully adjusted model fit is shown in Appendix A. 

Breast cancer risk was highest in current users of combined MHT (*n* = 2129) (HR 1.64, 95% CI 1.32–2.03), compared with never users. Risk amongst current users of oestrogen only was (*n* = 2278) (HR 1.03, 95% CI 0.79–1.34) compared with never users (Table 5).

Analysis was repeated for ER+ breast cancers only, which comprised 88% of breast cancer diagnoses in the cohort. Current MHT users (combined and oestrogen only) were at increased risk of an ER+ diagnosis BC (HR 1.45, 95% CI 1.20–1.74) (Appendix A).

### 3.4. Current BMI and MHT and Breast Cancer Risk

Current BMI was positively associated with breast cancer risk (adjusted HR 1.23 per 5 kg/m^2^, 95% CI 1.16 to 1.30; Appendix A). The effect of MHT in current users was attenuated by BMI (adjusted interaction 0.81, 95% CI 0.67 to 0.98; Appendix A). This is illustrated by the data in Table 1 Part A, where there is less difference in risk between never and current users of MHT in the higher BMI category than the lower BMI category.

### 3.5. Early Adulthood BMI and MHT and Breast Cancer Risk

BMI in young adulthood was inversely associated with breast cancer risk (adjusted HR 0.77 per 5 kg/m^2^, 95% CI 0.69 to 0.87; Appendix A). BMI at age 20 did not attenuate risk of current MHT use (adjusted interaction 1.05, 95% CI 0.72 to 1.53; Appendix A). Table 1 Part B shows high early BMI reduced risk across never, former and current users of MHT, MHT did however increase risk across women with early BMI above and below the median.

### 3.6. Combined Stratified Model with Current BMI, Early BMI and MHT and Breast Cancer Risk

Risk of breast cancer amongst MHT users was stratified for ≥ and < median for current BMI (26.4 kg/m^2^) and ≥ and < median for BMI age 20 years (21.6 kg/m^2^) (Table 1 Part C).

Higher BMI at age 20 appears to attenuate the effects of high current BMI on BC risk both amongst women who are not using MHT and amongst current MHT users (adjusted interaction BMI *BMI 20 HR 0.96, 95% CI 0.90–1.02; Appendix A). 

There is no specific interaction between MHT use, current BMI and early BMI (HR 0.99, 95% CI 0.74–1.31) (Appendix A). Risk was highest amongst MHT users with current BMI higher than the median and BMI at 20 less than the median (HR 1.84, 95% CI 1.22–2.76) (Table 1c). Analysis for ER+ breast cancer found comparable results to the overall analysis (Appendix A).

### 3.7. Effects of Oestrogen Only and Combined MHT and Current and Early BMI and Breast Cancer Risk

Current combined MHT use increased risk amongst women irrespective of their current and early adulthood BMI (Table 3 Part A). Oestrogen only MHT use did not significantly increase risk in current or former MHT users in any of the BMI groups (Table 3 parts A, B, and C). However, risk appeared lower amongst current users of oestrogen only MHT with early BMI > median (HR 0.67, 95% CI 1.45–1.00) compared to early BMI < median (HR 1.23, 95% CI 0.87–1.73) (Table 3 Part B). Attenuation of risk amongst current oestrogen only MHT users with current BMI > median is seen in women who also had early BMI > median (HR 0.75, 95% CI 0.44–1.28) compared to women with early BMI < median (HR 1.68, 95% CI 0.97–2.90) (Table 3 Part C).

### 3.8. MHT Use and Age at First Pregnancy

Women who had a late pregnancy had an increased risk of breast cancer (age of first pregnancy >35 years HR 1.38, 95% CI 1.08–1.76) (Appendix A). Late first pregnancy or nulliparity increased risk across MHT users and non-users (Table 2). There is no specific interaction between MHT use, age of pregnancy or nulliparity (HR 1.00, 95% CI 0.80–1.02) (Appendix A).

### 3.9. Effects of Oestrogen Only and Combined MHT and Age of First Pregnancy 

Risks were comparable with combined MHT use amongst women with age of first pregnancy <30 years (HR 1.74, 95% CI 1.34–2.25) and > age 30 or nulliparity (HR 1.90, 95% CI 1.16–3.13). However, risks appeared lower amongst current oestrogen only MHT users with age of first pregnancy <30 years (HR 0.96, 95% CI 0.71–1.31) compared to >age 30 or nulliparity (HR 1.81, 95% CI 0.93–3.53) (Table 2). 

## 4. Discussion

In this cohort, women with a lower current BMI were more likely to be current MHT users compared to women with a higher BMI. We have confirmed that current use of MHT increases breast cancer risk with excess risk mainly attributed to use of combined MHT. Higher early adulthood BMI had a small reduction in risk across never, former and current MHT groups. Late first pregnancy or nulliparity increased risk across never, former and current use of MHT groups. Neither BMI at age 20 or late first pregnancy or nulliparity had a specific modifying effect on the breast cancer risk related to overall MHT use. Observations of lower risks with oestrogen only MHT amongst women with high early BMI and early age of first pregnancy are interesting and require further study in larger cohorts.

Previous studies have reported that women with higher BMI were less likely to have used MHT [14,15,16]. Possible reasons for less use of MHT by heavier women include: experiencing fewer menopausal symptoms although this seems unlikely as the majority of papers report increased vasomotor symptoms in heavier women [17,18]; reduced likelihood of heavier women engaging with health behaviours, and contra-indications to MHT prescription associated with higher risk of thrombosis [19]. 

Our findings of increased breast cancer risk with current combined MHT, particularly with ER+ cancers concur with those found by the Collaborative Group [2]. We did not however observe an increased breast cancer risk for former MHT users. There was no significant association with oestrogen only MHT and BC risk, however our reported confidence intervals of the HR for oestrogen only MHT overlap with that reported by the Collaborative Group, indicating the results are broadly similar.

In these analyses we observed that higher current BMI attenuates the BC risk associated with overall current (combined and oestrogen only) MHT use. The attenuation of BC risk associated with oestrogen only MHT amongst currently heavier women has previously been reported [2]. Postmenopausal oestrogen levels correlate with BMI since endogenous oestrogen synthesis occurs within adipose tissue. The observed attenuation of oestrogen MHT risk is thought to reflect that exogenous oestrogen do not further stimulate the breast tissue in heavier women. This is consistent with a previous stated model that proposes a threshold for free oestrogen concentration beyond which there is no additional risk of breast cancer [20]. 

The stratified analysis in the current study suggests the attenuation that effects of oestrogen only MHT amongst currently heavier women is mainly seen in women who were heavier at an early age, and are not seen in formerly lighter women. The confidence intervals of these associations are quite wide due to small numbers of current users. However this observation raises the possibility that the apparent attenuation of oestrogen only MHT breast cancer risk amongst currently heavier women may be related to early rather than current BMI effects. Previous reports summarised in the collaborative overview did not examine the effects of early BMI.

A previous analysis within the PROCAS cohort reported that for women with early adulthood BMI >23.4 kg/m^2^ (top 25% centile) neither attained adult BMI nor adult weight gain was associated with breast cancer risk [8]. The observation that higher BMI in early adulthood attenuates the BC risk associated with later adiposity has been reported in a number of studies [21,22,23]. A significant part of BC risk associated with postmenopausal BMI is thought to be mediated by increased oestrogen levels and the associated stimulation of breast tissue proliferation [24]. Reduced breast tissue proliferation has been reported amongst pre and postmenopausal women who had been heavier at age 18 (BMI >22 kg/m^2^) [25]. Higher early adulthood BMI may attenuate the proliferative response of breast tissue to endogenous (associated with current BMI) or exogenous (MHT) oestrogen through a number of mechanisms. These include reducing terminal end buds and ductal elongation, and an overall reduction in the number of cells within breast tissue [25,26,27] and decreased expression of genes involved with both oestrogen action, i.e., ESR1 and GATA3, and cell proliferation, i.e., RPS6KB1, in breast tissue [28]. Also, increased levels of bioavailable oestrogens in early adulthood (associated with insulin levels and decreased sex hormone binding globulin) [29] could induce earlier differentiation of mammary cells [30] and expression of the BRCA1 tumour suppressor gene [31]. Higher levels of insulin-like growth factor 1 (IGF-1) during childhood and adolescence are associated with lower levels in adulthood [32]. The synergistic effects and cross talk between IGF-1 and oestrogen and their receptors are well established [33]. In addition women who are heavier in early adulthood are likely to have greater numbers of adipocytes (a hyperplastic phenotype). In contrast formerly lean women who gain weight will develop adipose hypertrophy (few but large cells) which are associated with inflammation and dysregulated metabolism [34].

A prospective study among 483,241 women and 7656 breast cancers studied whether MHT-associated BC risk is modified by life course patterns of BMI. The study reported that current users of MHT who reported being overweight at the ages of 7 and 15 (self-identified as being heavier than their peers) were at higher risk (HR 1.68, 95% CI 1.32–2.14) compared to never users of MHT who were overweight as young. However, risks were higher amongst current MHT users who remained at normal weight throughout adult life (HR 2.25, 95% CI 1.93–2.62) or who had gained weight (HR 2.28, 95% CI 1.94–2.67) [35]. The authors reported these risks were higher than expected when adding the separate risks of BMI and MHT with respective relative excess risk due to the interaction scores of 0.52 (95% CI 0.09–0.95) and 0.37 (95% CI -0.07 – +0.08). They concluded that women who were overweight at a young age were less susceptible to the effects of MHT than women who remained a normal weight or who gain weight in adulthood. This effect was seen amongst the whole cohort (70% combined MHT and 23% oestrogen only) and within the combined only group. They did not report the associations amongst current oestrogen only MHT users. The findings from this large cohort are of interest, however the analyses of early weight did not adjust for current BMI. Since higher weight at a younger age is likely to result in a higher current BMI, this study may not distinguish any interactions between early weight and current BMI and the effects of MHT. 

We found that late age at first pregnancy increased risk amongst never, current and former MHT users. Age at first pregnancy does not modify risk associated with overall MHT use and breast cancer. Oestrogen only MHT appeared to be associated with a greater risk amongst women with a late first pregnancy or nulliparity compared to women with age of first pregnancy <30. There was no modification of the risk of combined MHT. 

Risk of late pregnancy relates to a prolonged duration of undifferentiated state of the mammary tissue [36]. Synergistic effects of nulliparity and high postmenopausal BMI on BC risk in women aged >70 years have been reported [37]. Murrow et al recently reported that parity and high BMI amongst premenopausal women both decreased the oestrogen and progesterone responsiveness of the breast. These effects were associated with respective reductions in hormone signalling in hormone responsive luminal cells and reductions in the proportion of hormone responsive luminal cells within the mammary epithelium [10]. This data suggests a common pathway that could be shared between early pregnancy and high young age adiposity that is protective against breast cancer.

Our stratified analysis aimed to study the independent and combined interactions of current BMI, early BMI, MHT use and BC risk. Also whether these interactions differed with combined and oestrogen only MHT. Our sample size limits the ability to study all of these relationships with sufficient power. The observed trends in higher early weight and early pregnancy reducing the risk of oestrogen only MHT breast cancer requires further investigation in larger cohort or consortium studies. 

Strengths of this analysis are that it was conducted in a large UK population. Many confounders associated with breast cancer risk were taken into consideration. The independent effects of both current BMI and early adulthood BMI were elicited in the models by including BMI at two different time points. Additionally, detailed information regarding MHT use and type were collected and breast cancer diagnosis updated on a regular basis. Sensitivity analysis showed that the models were consistent. 

All information used in this analysis were self-reported, including current weight, height, weight at age 20, age at first pregnancy, name of MHT, how long MHT was used for and when MHT was stopped for former users. It is well known that there is a bias of underreporting weight and over reporting height [38]. However validation studies show self-reported BMI is highly correlated with independently measured weight and the mean difference between self-reported and measured weight is minimal [8,39]. BMI at entry and at age 20 was missing for 7.0% and 12.4% of the study population respectively. Recall bias could also occur for variables such as MHT duration. We did not update HRT usage status during follow up, and so were unable to estimate any association between duration of HRT use and risk of BC. The median value of the cohort was assumed where data was missing, and other methods could be used to account for the missing values. Breast cancer risk varies across racial groups [40] and the majority of women in the study were Caucasian thus limiting generalization of the findings to other ethnic groups. Risk factor information was only collected at baseline and it is possible that MHT status, BMI and other risk factors changed for some women. Some current MHT users are likely to have become former users and premenopausal women who had not used MHT could have become postmenopausal and started using MHT during the eight years of follow up. 

Implications for practice for this research include that clinical risk assessment of suitability of a woman to commence on MHT, should include consideration of their current BMI and potentially early adulthood BMI and the type of MHT to be prescribed. The findings of this study support recommendations to maintain a healthy weight across the life course. Breast cancer risk is similar among women with higher current BMI who never used MHT and women with lower BMI who use MHT. The smaller increase in risk with MHT amongst heavier women should not deter these women from losing weight. The highest BC risk is seen amongst current MHT users with a high current BMI, especially those with a low BMI at an early age. Women at increased weight taking MHT will have higher risks of other MHT associated adverse effects including venous thromboembolism, stroke and endometrial cancer [5]. Weight loss has been shown to decrease risk of other cancers including colorectal and endometrial cancer, and additionally might help manage menopausal symptoms [41].

Future studies need to investigate the associations between current and early BMI and age of pregnancy and MHT associated breast cancer risk. These studies could determine whether there are specific BMI ranges where these effects occur and whether the range is different within different ethnic groups.

## 5. Conclusions

Combined oestrogen and progestagen MHT was associated with the highest BC risk. This risk was not modified by early or current BMI and age of pregnancy. Exploratory analysis amongst oestrogen only MHT users showed an attenuation of risk with early BMI greater than or equal to the median compared to less than the median and with age of first pregnancy less than 30 years compared to equal or greater than age 30 or nulliparity which require further study. Identifying characteristics which modify a woman’s MHT associated BC risk will allow their individual risks and benefits to be assessed and appropriate prescription of MHT to manage troublesome menopausal symptoms.

## Figures and Tables

**Figure 1 cancers-13-02710-f001:**
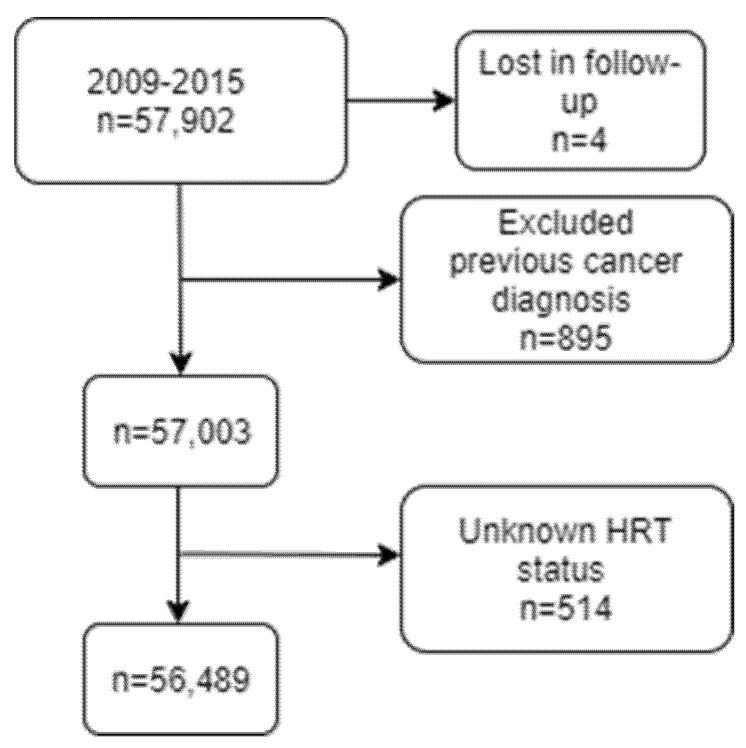
Flow diagram: Number of women in the cohort meeting the criteria for inclusion in the analysis.

**Table 1 cancers-13-02710-t001:** MHT and BC risk and current and early BMI.

**A**				
MHT use	**BMI < 26.4 kg/m^2^**		**BMI ≥ 26.4 kg/m^2^**	
	*N* (%) BC/ no BC	HR (95% CI)	*N* (%) BC/no BC	HR (95% CI)
Never	474 (2.5)/18,381 (97.5)	1.00 (Ref)	524 (3.2)/16016 (96.8)	1.44 (1.26–1.64)
Former	256 (3.2)/7858 (96.8)	1.12 (0.95–1.32)	251 (3.2)/7508 (96.8)	1.29 (1.09–1.53)
Current	101 (3.9)/2457 (96.1)	1.60 (1.28–1.98)	57 (3.2)/1725 (96.8)	1.45 (1.09–1.92)
**B**				
	**BMI age 20 < 21.6 kg/m^2^**		**BMI age 20 ≥ 21.6 kg/m^2^**	
	*N* (%) BC/noBC	HR (95% CI)	*N* (%)BC/no BC	HR (95% CI)
Never	439 (3.0)/14,292 (97.0)	1.00 (Ref)	559 (2.7)/20,105 (97.3)	0.85 (0.75–0.97)
Former	235 (3.3)/6878 (96.7)	0.98 (0.83–1.16)	272 (3.1)/8488	0.89 (0.75–1.05)
Current	80 (4.0)/1943 (96.0)	1.37 (1.08–1.75)	78 (3.4)/2239 (96.6)	1.11 (0.86–1.41)
**C**				
	**BMI age 20 < 21.6 kg/m^2^**		**BMI age 20 ≥ 21.6 kg/m^2^**	
	*N* (%) BC/no BC	HR (95% CI)	*N* (%) BC/no BC	HR (95% CI)
Never BMI < 26.4 kg/m^2^	259 (2.6)/9737 (97.4)	1.00 (Ref)	139 (2.2)/6171 (97.8)	0.86 (0.70–1.06)
Never BMI ≥ 26.4 kg/m^2^	180 (3.8)/4555 (96.2)	1.53 (1.26–1.85)	420 (2.9)/13,934 (97.1)	1.20 (1.02–1.40)
Former BMI < 26.4 kg/m^2^	142 (3.2)/4249 (96.8)	1.12 (0.90–1.38)	83 (3.0)/2726 (97.0)	1.03 (0.80–1.33)
Former BMI ≥ 26.4 kg/m^2^	93 (3.4)/2629 (96.6)	1.23 (0.97–1.57)	189 (3.2)/5762 (96.8)	1.17 (0.96–1.43)
Current BMI < 26.4 kg/m^2^	54 (3.8)/1385 (96.2)	1.48 (1.10–2.00)	34 (4.0)/808 (96.0)	1.60 (1.12–2.30)
Current BMI ≥ 26.4 kg/m^2^	558 (95.5)/26 (4.5)	1.84 (1.22–2.76)	1431 (97.0)/44 (3.0)	1.24 (0.90–1.71)

BC breast cancer; Units: age (1 year), BMI (5 BMI units), BMI at age 20 (5 BMI units), height (5 cm), age at menopause (1 year), exercise (1 h per week), alcohol (1 unit per week), age at menarche (1 year) BMI median: 26.4 kg/m^2^, BMI20 median: 21.6 kg/m.

**Table 2 cancers-13-02710-t002:** MHT status and BC risk and age of pregnancy > 30 and nulliparity.

	**Age at Pregnancy < 30**		**Age at Pregnancy ≥ 30**	
	*N* (%) BC/ no BC	HR (95% CI)	*N* (%) BC/ no BC	HR (95% CI)
Never	670 (2.7)/24,126 (97.3)	1.00 (Ref)	328 (3.1)/10,271 (96.9)	1.20 (1.05–1.38)
Former	391 (3.1)/12,401 (96.9)	0.99 (0.86–1.13)	116 (3.8)/2965 (96.2)	1.25 (1.02–1.53)
Current	110 (3.5)/3076 (96.5)	1.31 (1.06–1.60)	48 (4.2)/1106 (95.8)	1.64 (1.22–2.20)
Oestrogen only				
	**Age at Pregnancy < 30**		**Age at Pregnancy ≥ 30**	
	*N* (%) BC/ no BC	HR (95% CI)	*N* (%) BC/ no BC	HR (95% CI)
Never	670 (2.7)/24,126 (97.3)	1.00 (Ref)	182 (3.2)/5465 (96.8)	1.28 (1.09–1.51)
Former	151 (2.7)/5399 (97.3)	0.89 (0.73–1.09)	14 (3.5)/390 (96.5)	1.22 (0.71–2.09)
Current	46 (2.6)/1715 (97.4)	0.96 (0.71–1.31)	9 (4.6)/187 (95.4)	1.81 (0.93–3.53)
Combined				
	**Age at Pregnancy < 30**		**Age at Pregnancy ≥ 30**	
	*N* (%) no BC/BC	HR (95% CI)	*N* (%) no BC/BC	HR (95% CI)
Never	670 (2.7)24,126 (97.3)	1.00 (Ref)	182 (3.2)/5465 (96.8)	1.29 (1.09–1.52)
Former	240 (3.3)/7002 (96.7)	1.07 (0.92–1.26)	41 (4.0)/984 (96.0)	1.36 (0.99–1.87)
Current	64 (4.5)/1361 (95.5)	1.74 (1.34–2.25)	16 (4.6)/331 (95.4)	1.90 (1.16–3.13)

BC breast cancer; Units: age (1 year), BMI (5 units), BMI at age 20 (5 units), height (5 cm), age at menopause (1 year), exercise (1 h per week), alcohol (1 unit per week), age at menarche (1 year).

**Table 3 cancers-13-02710-t003:** MHT type and BC risk and current and early BMI.

**A**				
	**BMI < 26.4 kg/m^2^**		**BMI ≥ 26.4 kg/m^2^**	
	*N* (%) BC/ no BC	HR (95% CI)	*N* (%) BC/ no BC	HR (95% CI)
Never	439 (3.0)/14,292 (97.0)	1.00 (Ref)	559 (2.7)/20,105 (97.3)	1.48 (1.29–1.69)
Former oestrogen only	79 (2.9)/2664 (97.1)	1.06 (0.81–1.38)	26 (2.1)/1199 (97.9)	1.23 (0.98–1.54)
Former combined	156 (3.6)/4214 (96.4)	1.17 (0.96–1.42)	52 (4.8)/1040 (95.2)	1.40 (1.15–1.69)
Current oestrogen only	37 (4.8)/984 (95.2	1.29 (0.90–1.85)	112 (3.0)/3670 (97.0)	1.16 (0.79–1.70)
Current combined	43 (4.3)/959 (95.7)	1.86 (1.40–2.47)	160 (3.2)/4818 (96.8)	2.11 (1.53–2.92)
**B**				
	**BMI age 20 < 21.6 kg/m^2^**		**BMI age 20 ≥ 21.6 kg/m^2^**	
	*N* (%) BC / no BC	HR (95% CI)	*N* (%) BC/ no BC	HR (95% CI)
Never	439 (3.0)/14,292 (97.0)	1.00 (Ref)	559(2.7)/20,105 (97.3)	0.84 (0.74–0.96)
Former oestrogen only	79 (2.9)/2664 (97.1)	0.85 (0.66–1.10)	112 (3.0)/3670 (97.0)	0.83 (0.66–1.04)
Former combined	156 (3.6)/4214 (96.4)	1.04 (0.86–1.25)	160 (3.2)/4818 (96.8)	0.88 (0.73–1.06)
Current oestrogen only	37 (4.8)/984 (95.2)	1.23 (0.87–1.73)	26 (2.1)/1199 (97.9)	0.67 (0.45–1.00)
Current combined	43 (4.3)/959 (95.7)	1.47 (1.07–2.01)	52 (4.8)/1040 (95.2)	1.54 (1.15–2.06)
**C**				
Oestrogen only MHT	**BMI age 20 < 21.6 kg/m^2^**		**BMI age 20 ≥ 21.6 kg/m^2^**	
	*N* (%) BC/ no BC	HR (95% CI)	*N* (%) BC/ no BC	HR (95% CI)
Never BMI < 26.4 kg/m^2^	259 (2.6)9737 (97.4)	1.00 (Ref)	6171 (97.8)/139 (2.2)	0.86 (0.70–1.05)
Never BMI ≥ 26.4 kg/m^2^	180 (3.8)/4555 (96.2)	1.54 (1.27–1.86)	420 (2.9)/13,934 (97.1)	1.20 (1.02–1.41)
Former BMI < 26.4 kg/m^2^	45 (2.9)/1484 (97.1)	1.05 (0.76–1.47)	30 (2.8)/1031 (97.2)	1.00 (0.67–1.48)
Former BMI ≥ 26.4 kg/m^2^	34 (2.8)/1180 (97.2)	1.04 (0.72–1.50)	82 (3.0)/2639 (97.0)	1.15 (0.87–1.50)
Current BMI < 26.4 kg/m^2^	23 (3.4)/663 (96.6)	1.33 (0.86–2.05)	11 (2.7)/395 (97.3)	1.06 (0.58–1.95)
Current BMI ≥ 26.4 kg/m^2^	14 (4.2)/321 (95.8)	1.68 (0.97–2.90)	15 (1.8)/804 (98.2)	0.75 (0.44–1.28)
Combined MHT	**BMI age 20 < 21.6 kg/m^2^**		**BMI age 20 ≥ 21.6 kg/m^2^**	
	*N* (%) BC/ no BC	HR (95% CI)	*N* (%) BC/ no BC	HR (95% CI)
Never BMI < 26.4 kg/m^2^	259 (2.6)9737 (97.4)	1.00 (Ref)	6171 (97.8)/139 (2.2)	0.86 (0.70–1.05)
Never BMI ≥ 26.4 kg/m^2^	180 (3.8)/4555 (96.2)	1.52 (1.25–1.84)	420 (2.9)13,934 (97.1)	1.19 (1.01–1.39)
Former BMI < 26.4 kg/m^2^	97 (3.4)/2765 (96.6)	1.14 (0.90–1.45)	53 (3.0)/1695 (97.0)	1.04 (0.77–1.40)
Former BMI ≥ 26.4 kg/m^2^	59 (3.9)/1449 (96.1)	1.36 (1.02–1.82)	107 (3.3)/3123 (96.7)	1.16 (0.92–1.47)
Current BMI < 26.4 kg/m^2^	31 (4.1)/722 (95.9)	1.60 (1.10–2.32)	23 (5.3)/413 (94.7)	2.08 (1.35–3.18)
Current BMI ≥ 26.4 kg/m^2^	12 (4.8)/237 (95.2)	1.96 (1.10–3.51)	29 (4.4)/627 (95.6)	1.80 (1.22–2.65)

BC breast cancer; Units: age (1 year), BMI (5 units), BMI at age 20 (5 units), height (5 cm), age at menopause (1 year), exercise (1 h per week), alcohol (1 unit per week), age at menarche (1 year); BMI median: 26.4 kg/m^2^, BMI age 20 median: 21.6 kg/m^2^.

**Table 4 cancers-13-02710-t004:** Baseline Characteristics of the 56,489 women in the PROCAS cohort (2009–2015).

	Status of MHT Use			
	Total	Never	Former	Current
**Number of women (%)**	56,489 (100)	35,933 (63.6)	16,149 (28.6)	4407 (7.8)
Ethnicity				
White	53,571 (94.8)	33,695 (93.8)	15,634 (96.8)	4242 (96.3)
Other	2918 (5.2)	2238 (6.2)	515 (3.2)	165 (3.7)
Unknown	1799	1027	648	124
Age at study entry				
Median (IQR)	56.9 (51.6–63.6)	53.9 (50.7–60.8)	62.9 (58.3–66.6)	55.2 (51.5–60.6)
Mean (SD)	57.8 (7.0)	56.0 (6.7)	62.3 (5.8)	56.4 (6.0)
Menopausal status				
premenopausal and perimenopausal	18,764 (33.2)	16,742 (46.6)	821 (5.1)	1201 (27.3)
postmenopausal	37,725 (66.8)	19,191 (53.4)	15,328 (94.9)	3206 (72.7)
Unknown	2831	2673	76	82
Age at menopause				
Median (IQR)	50.0 (46.0–53.0)	50.0 (48.0–53.0)	50.0 (45.0–53.0)	48.0 (44.0–51.0)
Mean (SD)	49.0 (5.6)	49.9 (4.7)	48.1 (6.4)	47.1 (6.6)
Unknown	46 (0.1)	29 (0.1)	15 (0.1)	2 (<0.1)
Age at menarche				
Median (IQR)	13.0 (12.0–14.0)	13.0 (12.0–14.0)	13.0 (12.0–14.0)	13.0 (12.0–14.0)
Mean (SD)	12.9 (1.6)	12.9 (1.6)	12.8 (1.6)	12.9 (1.6)
Unknown	1186 (2.1)	762 (2.1)	319 (2.0)	105 (2.4)
Parity and age at first pregnancy				
Nulliparous	7330 (13.0)	5028 (14.0)	1681 (10.4)	621 (14.1)
<30	41,408 (73.3)	25,259 (70.2)	13,013 (18.5)	3236 (73.4)
≥30	7751 (13.7)	5706 (15.8)	1455 (2.2)	550 (12.5)
Median age at first pregnancy (IQR)	24.0 (21.0–28.0)	24.0 (21.0–28.0)	23.0 (20.0–26.0)	23.0 (20.0–27.0)
Mean age at first pregnancy (SD)	24.4 (5.3)	25 (5.3)	23.5 (4.6)	24.1 (5.2)
Unknown	4 (<0.1)	3 (<0.1)	1 (<0.1)	0 (<0.1)
BMI				
Underweight or normal weight (<25 kg/m^2^)	20,216 (35.9)	12,931 (36.1)	5486 (34.1)	1799 (41.0)
Overweight (25–29.9 kg/m^2^)	22,249 (39.5)	13,813 (38.5)	6660 (41.4)	1776 (40.4)
Obese (≥30 kg/m^2^)	13,868 (24.6)	9091 (25.4)	3959 (24.6)	818 (18.6)
Median (IQR)	26.4 (23.8–29.9)	26.4 (23.7–30.0)	26.4 (24.0–29.9)	26.0 (23.4–28.7)
Mean (SD)	27.4 (5.4)	27.5 (5.5)	27.4 (5.1)	26.5 (4.7)
Unknown	3801	2587	933	281
BMI at age 20				
Underweight or normal weight (<25 kg/m^2^)	50,119 (88.8)	31,673 (88.2)	14,456 (89.6)	3990 (90.6)
Overweight (25–29.9 kg/m^2^)	5105 (9.0)	3369 (9.4)	1388 (8.6)	348 (7.9)
Obese (≥30 kg/m^2^)	1243 (2.2)	877 (2.4)	298 (1.8)	68 (1.5)
Median (IQR)	21.6 (20.3–23.0)	21.6 (20.3–23.2)	21.6 (20.2–23.0)	21.6 (20.0–22.8)
Mean (SD)	22.0 (3.1)	22.1 (3.1)	21.9 (2.9)	21.8 (2.9)
Unknown	6777	4694	1600	483
Mean Height (SD)	1.62 (0.07)	1.62 (0.07)	1.61 (0.06)	1.62 (0.06)
Median Height (IQR)	1.63 (1.57–1.65)	1.63 (1.57–1.65)	1.60 (1.57–1.65)	1.63 (1.57–1.68)
Unknown	1111	766	269	76
Oophorectomy				
Yes	6696 (12.0)	1999 (5.6)	3584 (22.2)	1213 (27.5)
No	49,693 (88.0)	33,934 (94.4)	12,565 (77.8)	3194 (72.5)
Unknown	8709	5183	2716	810
MHT type				
Oestrogen only	8895 (43.2)		6617 (41.0)	2278 (51.7)
Combined	11,661 (56.7)		9532 (59.0)	2129 (48.3)
Unknown	13,456		12,441	1015
Duration of MHT use (years)				
Μedian (IQR)	5 (2.0–10.0)		5 (2.0–10.0)	7 (2.5–13.0)
Mean (SD)	7.3 (4.6)		6.2 (5.4)	8.6 (7.2)
Unknown	2433		1178	1255
VAS Density				
Median (IQR)	24.8 (14.9–35.8)	24.8 (15.6–35.8)	22.4 (12.9–34.4)	29.5 (18.9–41.0)
Mean (SD)	27.1 (16.2)	27.5 (16.1)	25.3 (15.8)	31.3 (16.8)
Family History of Breast and/ or Ovarian Cancer *				
Yes	15,738 (27.9)	10,185 (28.3)	4379 (27.1)	1174 (26.6)
No	40,748 (72.1)	25,748 (71.7)	11,768 (72.9)	3232 (73.4)
Median exercise hours per week (IQR)	3.5 (1.5–7.0)	3.5 (1.0–6.0)	3.5 (2.0–8.0)	3.5 (2.0–7.0)
Unknown	17,085	11,171	4630	1284
Median alcohol units per week (IQR)	4.0 (0.0–10.0)	4.0 (0.0–10.0)	4.0 (0.0–10.0)	4.0 (1.0–10.0)
Unknown	2512	1593	723	196
Median ΕΙΜD score 2010 (IQR) **	18.9 (10.4–35.1)	19.4 (10.5–35.4)	18.7 (10.6–34.9)	17.6 (9.7–32.5)
Unknown	312	176	111	25

BMI body mass index, SD standard deviation, IQR interquartile range; MHT hormonal replacement therapy, VAS visual analogue scale; EIMD English Index of multiple deprivation. * First or second degree relative with ovarian and/ or breast cancer; ** Ever vs. never MHT EIMD 2010 *p* = 0.002, EIMD current vs. never *p* < 0.001, never vs. former *p* = 0.444; age adjusted comparisons: BMI current vs. never *p* < 0.001, never vs. former *p* = 0.639; age adjusted VAS current vs. never *p* < 0.001, never vs. former *p* = 0.697; age, menopausal status, BMI adjusted VAS current vs. never *p* < 0.001, never vs. former *p* = 0.001.

**Table 5 cancers-13-02710-t005:** MHT status and type of MHT and breast cancer risk.

MHT Use Status	*N*% BC/ N% no BC	HR (95% CI)	*p*-Value	HR (95% CI)	*p*-Value
Never	998 (2.8)/ 34,397 (97.2)	1.00		1.00	
Former	507 (3.2)/ 15,366 (96.8)	0.98 (0.88–1.10)	0.765	1.03 (0.91–1.17)	0.614
Current	158 (3.6)/ 4182 (96.4)	1.27 (1.07–1.50)	0.006	1.35 (1.13–1.60)	0.001
**MHT type**					
Never	998 (2.8)/ 34,397 (97.2)	1.00		1.00	
Former oestrogen only	191 (2.9)/ 6334 (97.1)	0.90 (0.76–1.05)	0.189	0.95 (0.79–1.14)	0.558
Former combined	316 (3.4)/ 9032 (96.6)	1.04 (0.91–1.19)	0.586	1.06 (0.93–1.22)	0.398
Current oestrogen only	63 (2.8)/ 2183 (97.2)	0.96 (0.74–1.24)	0.754	1.03 (0.79–1.34)	0.835
Current combined	95 (4.5)/ 1999 (95.5)	1.60 (1.30–1.98)	<0.001	1.64 (1.32–2.03)	<0.001

BC breast cancer, HR hazard ratio, CI confidence interval. Fully adjusted for age at consent (1 year), BMI (5 units), BMI at age 20 (5 units), height (5 cm), age at menarche (1 year), age at menopause (1 year), menopausal status (pre/perimenopausal vs. postmenopausal), ethnicity (white vs. other), alcohol consumption (1 unit/week), exercise (1 h/week), age at first pregnancy (<20, 20–24, 25–29, 30–34, ≥35), oophorectomy (yes vs. no), family history (yes vs. no).

## Data Availability

The dataset used and analysed during the current study is available from the corresponding author on reasonable request.

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
