# Peer review of "Is Breast Cancer Risk Associated with Menopausal Hormone Therapy Modified by Current or Early Adulthood BMI or Age of First Pregnancy?"

_cancers, 2021, doi:10.3390/cancers13112710_

Round 1

Reviewer 1 Report

The current study is on a topic of relevance and general interest to the readers of the journal. I found the paper to be overall well written and felt confident that the authors performed careful and thorough analysis.

Do the authors think that regular mammographic screening can help this specific population as there is an understanding that increase in breast density  is caused by hormonal treatment. 

Can the authors sum up what the findings mean in the “grand scheme of things”?

Author Response

The current study is on a topic of relevance and general interest to the readers of the journal. I found the paper to be overall well written and felt confident that the authors performed careful and thorough analysis.

Do the authors think that regular mammographic screening can help this specific population as there is an understanding that increase in breast density is caused by hormonal treatment. 

The study was conducted amongst a screening population. The issue of sensitivity of screening amongst HRT users is an interesting point but not discussed as not directly relevant to the analysis

Can the authors sum up what the findings mean in the “grand scheme of things”?

We have discussed implications in the conclusions sections.. Principally that Implications for practice for this research include that clinical risk assessment of suitability of a woman to commence on MHT, should include consideration of their current BMI and potentially early adulthood BMI and the type of MHT to be prescribed

Reviewer 2 Report

Is breast cancer risk associated with menopausal hormone therapy modified by current or early adulthood BMI or age of first pregnancy?   

This is an interesting manuscript to published in cancers and however it needs  more additional analysis 

Minor comments: Its is complicated story and it could be simplified 

Major comments: This article is build based on observational evidences including age, pregnancy time , ethnicity. It will be nice to add biomarkers from blood serum and compare with age and  first pregnancy. These pare meters will enhance the quality of the manuscript and will invite broad readers.

Author Response

Is breast cancer risk associated with menopausal hormone therapy modified by current or early adulthood BMI or age of first pregnancy?  

This is an interesting manuscript to published in cancers and however it needs more additional analysis 

Minor comments: Its is complicated story and it could be simplified 

We thank the reviewer for these suggestions. None of the other 3 reviewers requested a re- write to simplify the paper so , we have not done this.

Major comments: This article is build based on observational evidences including age, pregnancy time , ethnicity. It will be nice to add biomarkers from blood serum and compare with age and first pregnancy. These pare meters will enhance the quality of the manuscript and will invite broad readers.

We accept there are limitations to observational research and that biomarker data would enhance the study. However the data is from an observational study which does not have biomarker data , so we are unable to fulfil this request

Reviewer 3 Report

This manuscript investigated the possibility that body weight at age 20 or the age at 1st pregnancy affected the relationship between hormone therapy and breast cancer risk in normal weight and overweight/obese women. The study confirmed that obese postmenopausal women are less affected by HT than lean women, but also showed that overweight and obesity without HT increase breast cancer risk at approximately equal amount than HT in lean women. The study also suggests that HT is less harmful in women who were not very slim (BMI<21.6) at the age of 20, which is an interesting observation.

Minor comments:

Page 4. Add to the text the % of women in the cohort who had their first child after age 30. This is provided in the Table, but should also be in the text.

Page 5. Add p-values for all differences reported in the text in the 1st and 2nd paragraph.

Page 5. The following sentence is unclear “Risk was lower amongst current users of oestrogen only (n=2278) (HR 1.03, 95% CI 0.79-1.34) compared with never users (Table 2).” The sentence seems to suggest that estrogen use lowered the risk, compared with never users. However, HR value suggests that the risk was not elevated. Please clarify.

Page 6. In the text, a reference to Table 3 d is made “Analysis for ER+ breast cancer found comparable results to the overall analysis (Table 3d).” However, no Table 3d is shown.

Page 9. Most of the paragraph speculating how adiposity during childhood and early adolescence may protect against later development of breast cancer should be deleted. These speculations are not relevant for this epidemiological study.

Author Response

Page 4. Add to the text the % of women in the cohort who had their first child after age 30. This is provided in the Table, but should also be in the text.

 26.7% of women had their first pregnancy either at or after the age of 30 years. This figure has been added to section 2.2

Page 5. Add p-values for all differences reported in the text in the 1st and 2nd paragraph.

All differences cited have P<0.001 due to large sample size, this statement has been added to the text for clarity.

Page 5. The following sentence is unclear “Risk was lower amongst current users of oestrogen only (n=2278) (HR 1.03, 95% CI 0.79-1.34) compared with never users (Table 2).” The sentence seems to suggest that estrogen use lowered the risk, compared with never users. However, HR value suggests that the risk was not elevated. Please clarify.

We have rewritten this sentence to clarify the actual OR of BC amongst current users of oestrogen only HRT on page 5

Page 6. In the text, a reference to Table 3 d is made “Analysis for ER+ breast cancer found comparable results to the overall analysis (Table 3d).” However, no Table 3d is shown.

 This has been corrected to supplementary table 3d

Page 9. Most of the paragraph speculating how adiposity during childhood and early adolescence may protect against later development of breast cancer should be deleted. These speculations are not relevant for this epidemiological study.

We would prefer to include this discussion and feel it is relevant to the analysis. This discussion is reviewing potential mechanisms whereby early weight could attenuate the proliferative effect of HRT, particularly oestrogen only HRT on breast tissue and hence a potential interaction between early weight, HRT use and BC risk

Reviewer 4 Report

congratulations on the manuscript and the great work. Here are my suggestions:

  •  Page 3: "N=3801 …." I suggest changing it:    3801 women….
  • Page 3: former user of MHT: For how long were they taking MHT? In the introduction, you pointed out, that the duration of MHT is significantly associated with BC occurrence. If you have no data, it has to be mentioned as a drawback at the end of the discussion.
  • As the weight at age 20 was recalled and reported by women in Questionnaires, this number is a rough assessment of real weight. As we cannot be sure, that these numbers are correct, the evidence is not very convincing. This needs to be mention in the manuscript as a drawback.

  • “…of risk with early BMI> median compared to <median and with age of first pregnancy <30 years compared to > age 30 or nulliparity which require further study,”  I suggest changing all marks in the text into the words.

Author Response

Page 3: "N=3801 …." I suggest changing it:  3801 women….

This has been amended as suggested

Page 3: former user of MHT: For how long were they taking MHT? In the introduction, you pointed out, that the duration of MHT is significantly associated with BC occurrence. If you have no data, it has to be mentioned as a drawback at the end of the discussion.

Duration of HRT use is reported for current and former users of HRT in table 1.

However we have acknowledged that we did not update HRT usage status during follow up. This meant we were unable to estimate any association between duration of HRT use and risk of BC. This point has been included in the limitation section of the discussion in the final paragraph of page 10

As the weight at age 20 was recalled and reported by women in Questionnaires, this number is a rough assessment of real weight. As we cannot be sure, that these numbers are correct, the evidence is not very convincing. This needs to be mention in the manuscript as a drawback.

This has been discussed in the discussion in the final paragraph of page 10

of risk with early BMI> median compared to <median and with age of first pregnancy <30 years compared to > age 30 or nulliparity which require further study,” I suggest changing all marks in the text into the words.

This has been changed as suggested on page 13

Round 2

Reviewer 2 Report

The authors did not respond any of my comments.